# Evolving Southern Ocean overturning in warming climates

Tingting Zhu ⓘ ✉ & Wei Liu ⓘ

The Southern Ocean Meridional Overturning Circulation (MOC) has intensified in recent decades, yet the interplay between its Eulerian and eddy components under future warming remains uncertain. Using ensemble climate simulations, here we show that the Eulerian-mean MOC shifts poleward under high-emission scenarios during the twenty-first century, with compensating eddy-induced MOC sustaining a uniformly intensified residual overturning. This response is less pronounced under low-emission scenarios with climate mitigation. Likewise, a poleward-shifted Eulerian-mean MOC occurred during the Mid-Pliocene Warm Period, but with weaker, broader eddy compensation, leading to non-uniform intensified residual overturning. Across past and future warming climates, the eddy-induced MOC is primarily modulated by surface heat flux changes at lower latitudes and by freshwater flux changes at higher latitudes over the Southern Ocean. The buoyancy forcing changes drive northward Antarctic upwelling, promoting Antarctic bottom water formation. Along with MOC changes, ventilation intensifies in the lower latitudes of the Southern Ocean as related to the subduction branch, especially during the past warm period.

The Southern Ocean plays a crucial role in the global climate system, particularly in the context of the meridional overturning circulation (MOC). Facilitating the upwelling of deep waters and the subduction of surface waters, the Southern Ocean MOC mediates the exchange between the deep ocean and the surface, thereby exerting a profound influence on regional or global climate[1,2]. A comprehensive understanding of the Southern Ocean MOC's past, present, and future behavior is essential for unraveling climate change in the 21st century.

The Southern Ocean MOC consists of Eulerian-mean and eddy-induced components, which counterbalance each other to generate a residual circulation. The Eulerian-mean component represents a time-mean clockwise MOC (positive in stream-function) while the eddy-induced MOC manifests a time-varying anticlockwise MOC (negative in stream-function) resulting from transient mesoscale eddies in the Southern Ocean. Anthropogenic forcing agents affect Southern Ocean MOCs by changing surface wind stress and buoyancy fluxes[3,4]. The overlying westerly winds drive the Eulerian-mean MOC, deepening isopycnals; this deepening, in turn, stimulates the eddy-induced circulation to act in a compensatory manner by flattening them[5]. In a changing climate, the altered eddy-induced MOC can partially compensate for changes in the Eulerian-mean MOC produced by Southern Hemisphere westerly wind shift and intensification[1]. However, the degree of such compensation appears sensitive to surface buoyancy conditions[6–8]. For instance, the surface buoyancy effect caused by a quadruple carbon dioxide rise can weaken the eddy-induced MOC, making it not compensate for changes in the Eulerian-mean MOC and thus helping to strengthen the residual MOC[9]. Under the Representative Concentration Pathway 8.5 (RCP8.5), interactions between Eulerian and eddy components of the MOC were suggested to be complex in climate models from the Coupled Model Intercomparison Project Phase 5 (CMIP5) and lack a uniform conclusion[10]. As such, eddy-induced and Eulerian-mean MOC responses, and the complex interaction and compensation between them, are still uncertain under future climate scenarios. These uncertainties potentially hinder our ability to accurately predict the Southern Ocean's contribution to future climate projections.

To address the above scientific question, we resort to ensemble simulations with CMIP6 climate models[11,12], and analyze the responses

Department of Earth and Planetary Sciences, University of California, Riverside, CA, USA. ✉e-mail: tzhu@ucr.edu

of the Southern Ocean's Eulerian-mean and eddy-induced MOCs to various climate forcings, with a particular focus on the physical mechanisms driving these responses. We will delve into a future scenario called the Shared Socioeconomic Pathway 5-85 (SSP585), which is an updated version of the RCP85 and describes a high-emission scenario with minimal efforts toward climate mitigation. We also look at the Mid-Pliocene Warm Period (mPWP; 3–3.2 Ma before present), which shares several key features with the projected future climate, including higher atmospheric carbon dioxide concentrations than preindustrial times, near-modern paleogeography, and elevated global temperatures[13]. In addition to the CMIP6 models, we will include two CESM1 models given the availability of ideal age tracer in the outputs of their mPWP simulations (Methods). By comparing future warming scenarios with past warm periods, we are able to assess the Southern Ocean's response to climate change, from the past to the future, and its role in Earth's system.

## Results

### Recent trend in the Southern Ocean MOC

Seen from observationally constrained estimate data (Methods), Southern Hemisphere surface westerly wind stress has shifted poleward and intensified from 1992 to 2017, with a maximum increasing trend exceeding 0.004 N/m$^2$ per decade at around 50°S (Fig. 1a, e). Over the Southern Ocean, the Eulerian-mean MOC exhibits a strengthening trend to the south of 40°S and a weakening trend to the north (Fig. 1b), with the weakening trend being particularly significant. Such a dipole pattern indicates a poleward shift and intensification of this time-mean circulation, driven by the displacement of surface westerly wind stress in response to rising greenhouse gas concentrations and stratospheric ozone depletion[14–18]. When the poleward-intensified winds strengthen and displace the Eulerian-mean MOC[4,19,20], the eddy-induced MOC presents an intensified trend on the southern side of 40°S, partially compensating for the trend in the Eulerian-mean MOC (Fig. 1c). Together, these trends contribute to the observed poleward shift and intensification of the residual MOC (Fig. 1d), with the intensification being particularly strong between 40°S and 60°S owing primarily to wind-driven circulation change. In addition to wind change, surface buoyancy forcing also exhibits notable variations. Surface freshwater flux increases outside the Antarctic, whilst surface heat flux alters in a non-uniform pattern (Fig. 1g, f), indicative of spatially varying contributions to changes in ocean stratification and MOC dynamics.

### Future change in the Southern Ocean MOC

The poleward-intensified Southern Ocean MOC during the recent decades highlights the need to understand future MOC change and underlying mechanisms by wind and buoyancy forcings. Under the SSP585 scenario, Southern Hemisphere westerly winds become stronger and displaces poleward by the end of the twenty-first century when compared to the end of the twentieth century (Fig. 2a). They strengthen the Eulerian-mean MOC in the southern region around 60°S and weaken it in the northern region around 40°S (Fig. 2b). The poleward-intensified Eulerian-mean MOC is partially offset by an eddy-induced MOC (Fig. 2c). In particular, the decelerated eddy-induced MOC near 40°S provides strong compensation for the weakened Eulerian-mean MOC in the same region, even surpassing it. This is especially pertinent to the varying surface buoyancy forcing, which combines changes in surface freshwater and heat fluxes, modulates Southern Ocean density structure and baroclinicity, and, in particular, reduces the isopycnal slope around 40°S (Fig. 1d), slowing the eddy-induced MOC there (Methods). Nonetheless, the enhanced eddy-induced MOC near 60°S offers a smaller compensation (Fig. 2d), thus the residual MOC predominantly alters with the Eulerian-mean MOC. Our findings demonstrate the compensating effect of eddy-induced MOC under future warming, which contrasts with previous findings[10].

We further probe the changes in surface heat and freshwater, or their combined thermal and haline effects, on the change in surface density flux. Relative to the historical period, surface density flux shows positive anomalies from 38°S to 45°S but negative anomalies to the south of 45°S under the SSP585 scenario (Fig. 3a–c), with the maximum decrease at around 55°S (Fig. 3d). Between the positive and negative, the altered surface density flux makes lower-latitude water denser while high-latitude water lighter (Fig. 3d), thus diminishing the isopycnal slope in this region (Fig. 2e). The augment and decline in surface density flux from 38°S to 45°S and from 45°S to 62°S are primarily due to reduced and enhanced downward net surface heat fluxes, a majority of which come from enlarged and diminished latent and sensible heat losses in these regions (Supplementary Fig. 1). The decrease in surface density flux to the south of 62°S is predominantly contributable to an increase in net surface freshwater input from melts of sea ice, icebergs, and ice shelves (Supplementary Fig. 2). It should be noted that surface freshwater flux change around Antarctica in CMIP6 models may be subject to uncertainty, as most models do not explicitly simulate Antarctic icesheet melting processes[21].

Surface density flux driven by heat and freshwater fluxes plays a key role in water mass transformation and formation[22] (Methods). Under the SSP585 scenario, the decrease in density flux, as mentioned before, results in a lighter surface density compared to the historical period. This is reflected in an extended area for density ranges between 23.5 kg/m$^3$ and 25.5 kg/m$^3$ and a contracted area for densities between 25.5 kg/m$^3$ and 28 kg/m$^3$ (Fig. 4a). The lightening transformation due to surface heat and freshwater fluxes has shifted toward lighter surface densities by the end of the twenty-first century (Fig. 4b, c). This shift suggests that the transformation toward lighter densities has extended further north. As a result, the total water mass transformation rate depicts an enhanced transformation toward lighter surface densities, with up to 35 Sv ($1 Sv = 10^6 m^3/s$) migrating from a density of approximately 25.7 kg/m$^3$ to lighter densities (black line in Fig. 4d).

When water masses accumulate or abate in volume within a given potential density range, the imbalance must be resolved through downwelling or upwelling. In the SSP585 scenario, the northward shift of the lightening transformation also causes the upwelling and downwelling regions to move northward (red and blue lines in Fig. 4e). Due to this movement, downwelling anomalies occur at densities below 25.7 kg/m$^3$, peaking at 8 Sv, while upwelling anomalies occur at densities above 25.7 kg/m$^3$, with a peak value of 6 Sv (black line in Fig. 4e). Besides, subduction anomalies appear around Antarctica, particularly in the Ross Sea and Weddell Sea, which are prompted by a northward displacement of upwelling and thereby support the bottom water mass formation in these regions (Fig. 4e, f), despite the fact that uncertainty may remain regarding how different models simulate water mass formation. The upwelling south of 50°S and downwelling north of 50°S help explain the diminished eddy-induced MOC response under the SSP585 (Fig. 2c).

Aside from the SSP585 high-emission scenario, we also look into the transition from the SSP126 low-emission scenario. By the end of the twenty-first century, Southern Hemisphere westerly winds, Southern Ocean Eulerian-mean, eddy-induced, and residual MOCs alter to a lesser extent under the SSP126 (Supplementary Fig. 3). Although the Eulerian-mean and eddy-induced MOCs compensate in slightly different ways, they result in a poleward-intensified residual MOC (Supplementary Fig. 3d) that follows a similar pattern to the SSP585 scenario (Fig. 2d). Our finding suggests that climate protection measures can diminish future changes in Southern Ocean winds and MOCs.

### Past change in the Southern Ocean MOC

The mPWP is widely considered a potential analog for near-future climate conditions, with atmospheric carbon dioxide concentrations

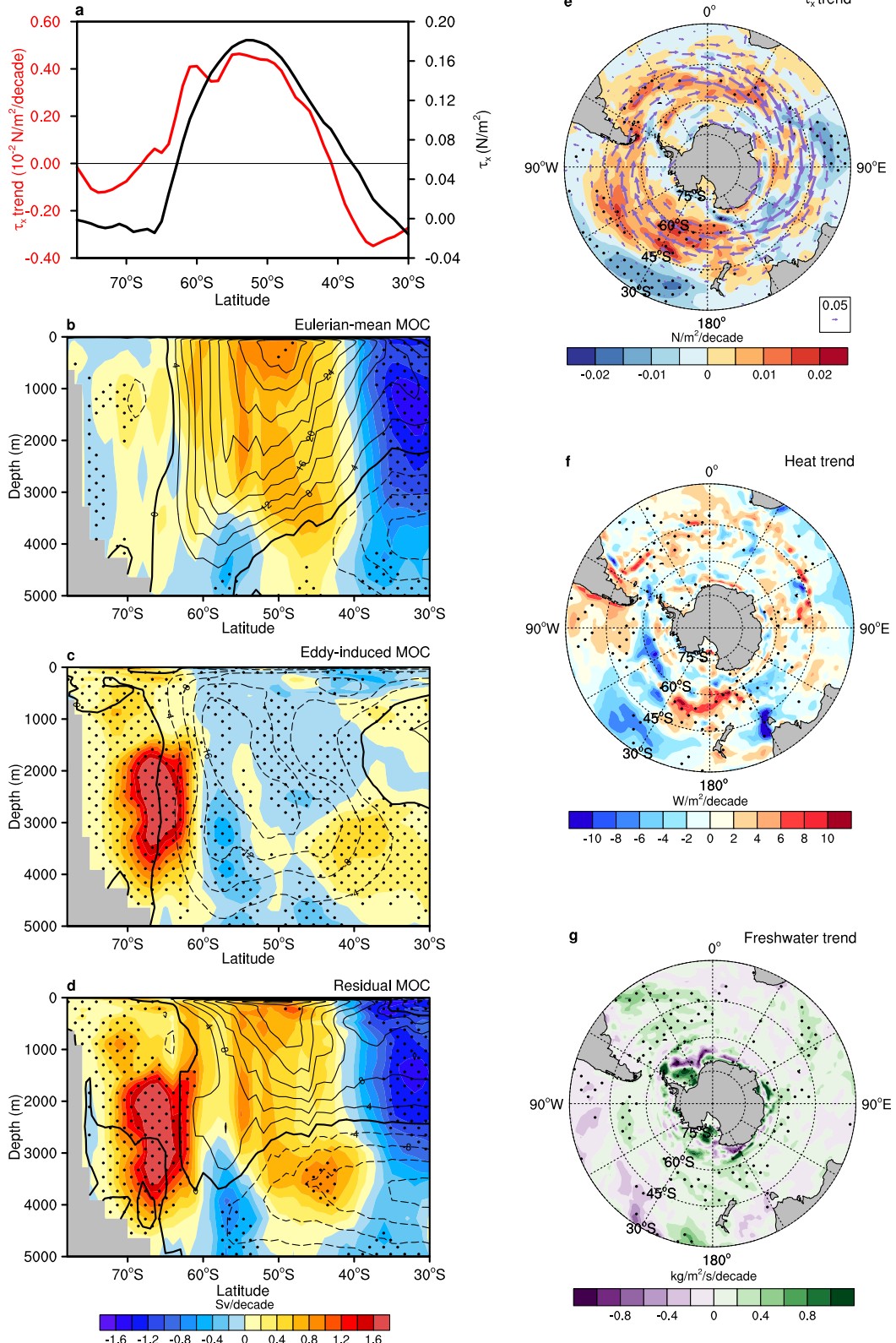

**Fig. 1 | Observed trends of Southern Ocean wind stress, meridional overturning circulation (MOC), heat and freshwater fluxes. a** Trend of annual and zonal mean surface zonal wind stress (red, left $y$-axis, units: $10^{-2}$ N/m²/decade) over the Southern Ocean overlaid with the climatological annual and zonal mean surface zonal wind stress (black, right $y$-axis, units: N/m²) from Estimating the Circulation and Climate of the Ocean Version 4 Release 4 (ECCO v4r4) between 1992 and 2017. **b** Trend (shaded, units: Sv) and climatology (contours, units: Sv/decade) of annual mean Southern Ocean Eulerian-mean MOC. **c** Same as **b**, but for the eddy-induced MOC. **d** Same as **b**, but for the residual MOC. **e** Trend of annual mean surface wind stress (shaded, units: N/m²) with annual mean climatological surface wind stress (vectors, units: N/m²) over the Southern Ocean. **f, g** Trends in annual mean surface heat and freshwater fluxes, respectively. Negative contours are dashed in (**b**–**d**). Dots indicate regions where trends are statistically significant at the 90% confidence level, determined using a two-tailed Student's $t$-test.

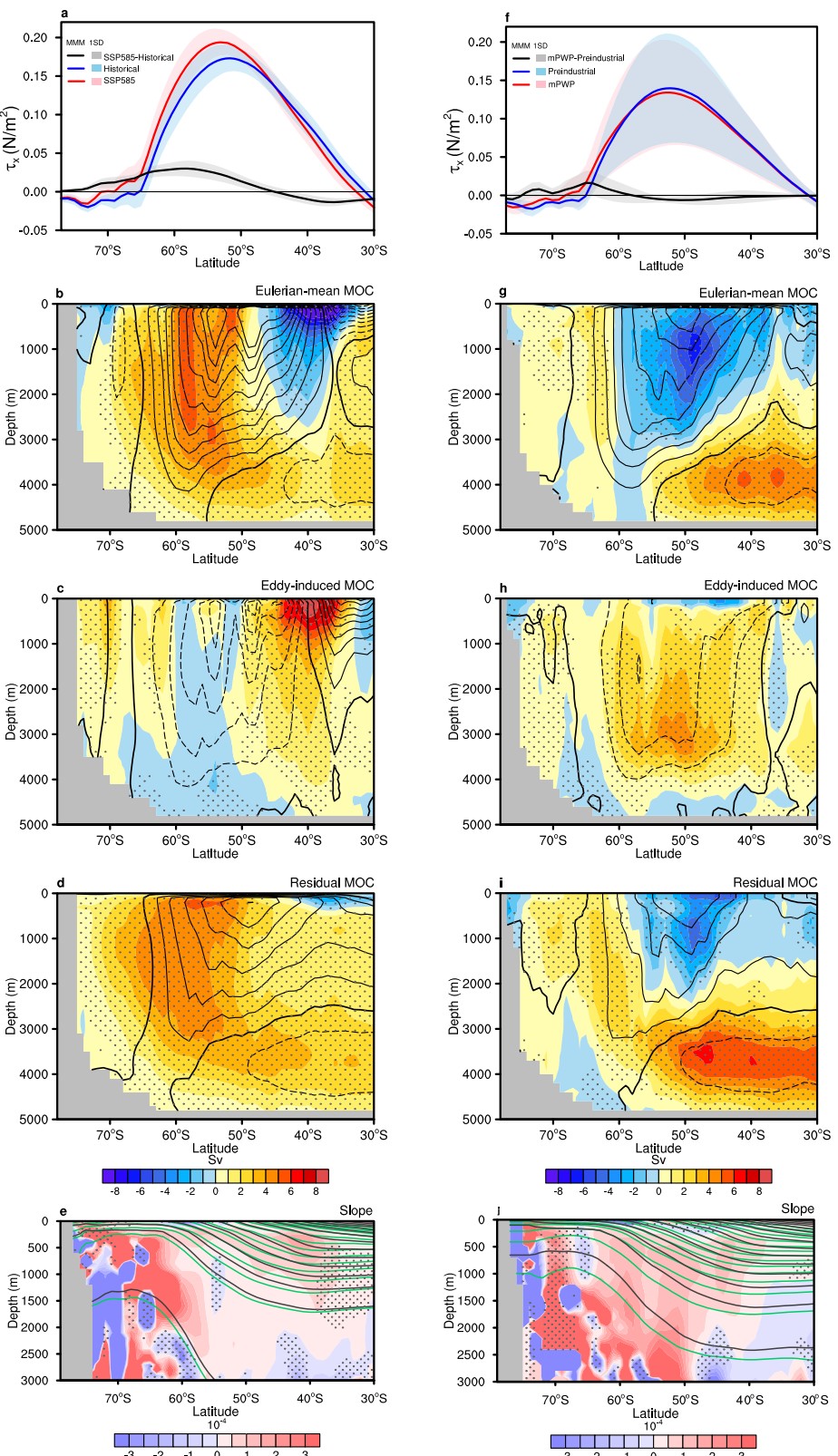

around 400 ppm−closely aligning with a warmer climate than pre-industrial times[13]. During this period, the global configuration of continents and ocean basins closely resembled the present day[23] while the Antarctic ice sheet was largely reduced. Global temperatures were about 2−3 °C warmer on average than the modern climate, comparable to projections for the end of the current century[24]. However, this warming was not uniform: high latitudes experienced pronounced

warming due to enhanced meridional ocean heat transport, strengthened thermohaline circulation, and dwindled sea ice[25,26], while tropical temperatures showed minimal differences from modern climate, as supported by paleoclimate reconstructions[27–29].

Relative to preindustrial times, the mPWP Eulerian-mean MOC witnesses a poleward shift, evidenced by an asymmetrical dipole pattern with a broad and strong weakening centered around 50°S and a small

**Fig. 2 | Changes in Southern Ocean meridional overturning circulations (MOC), zonal wind stress, and isopycnal slope in the Shared Socioeconomic Pathway 5–8.5 (SSP585) and the mid-Pliocene Warm Period (mPWP). a** Annual and zonal mean surface zonal wind stress in CMIP5/6 SSP585 [multi-model mean (MMM), red; inter-model spread, light red], historical (MMM, blue; inter-model spread, light blue) simulations, and their difference (SSP585 minus historical, MMM, black; inter-model spread, gray) over the Southern Ocean. The inter-model spread is defined as one standard deviation among models. **b** Difference in annual mean Southern Ocean Eulerian-mean MOC (shaded, units: Sv) for the MMM between SSP585 and

historical simulations, overlaid with historical Eulerian-mean MOC annual mean climatology (contours, units: Sv). **c** Same as **b**, but for the eddy-induced MOC. **d** Same as **b**, but for the residual MOC. **e** Annual mean potential density (green contours, SSP585; black contours, historical) and the difference of the slope of annual mean buoyancy surfaces (shaded, SSP585 minus historical) in the Southern Ocean. **f–j** Same as **a–e** but for the mPWP and preindustrial control as well as their difference. The stipples refer to the regions where at least two-thirds of the models agree with the sign of the difference in the multi-model mean.

intensification near 65°S (Fig. 2g). This shift is driven by Southern Hemisphere westerly winds, which have also displaced poleward, similar to the SSP585 poleward wind displacement (Fig. 2a, f). However, it is important to note that the westerly winds slightly weaken during the mPWP, in contrast to the wind intensification under the SSP585 future scenario. Such weakened mPWP winds can be attributed to changes in thermal gradients and ice volume[30], in that ice sheet retreat amplifies polar warming, diminishing the equator-to-pole temperature gradient and relaxing westerly winds[31,32]. As a result, the Eulerian-mean MOC delineates a dipole-like change, with a strong deceleration north of 60°S and a much weaker acceleration to the south.

Meanwhile, isopycnals flatten during the mPWP when compared to preindustrial times, engendering positive anomalies of isopycnal slope (Fig. 2j) that extend much broader and deeper than those from the historical to SSP585 scenario (Fig. 2e). This weakens the eddy-induced MOC over the Southern Ocean in a board way (Fig. 2h). Alterations in mPWP density reflect a combined effect of ice cover and carbon dioxide, both of which lead to a warmer, fresher surface water, stronger stratification in the Southern Ocean[33]. Nevertheless, the eddy-induced MOC weakening is not enough to fully compensate for the slowdown of the Eulerian-mean MOC around 50°S (Fig. 2h). As a result, the residual MOC closely resembles the Eulerian-mean MOC, altering in a dipole-like manner but with opposite changes to the north and south of around 56°S (Fig. 2i), which is at odds with the uniform intensification in the future warming scenario.

The distinct MOC variations between the mPWP and SSP585 also relate to distinct changes in surface density flux. In contrast to pre-industrial times, surface density flux generally decreases and increases to the south and north of around 48°S (Supplementary Fig. 4a–d) during the mPWP. Particularly, surface density flux change is primarily thermally driven to the north of 62°S. To the south, the surface freshening effect, which is most likely caused by sea ice, icebergs, and ice shelf melts, dominates the thermal effect, leading to a significant drop in the surface density flux (Supplementary Fig. 4d–f). Compared to the future warming scenario, surface density flux change during the mPWP has a smaller magnitude over the Southern Ocean, except for that close to Antarctica. It manifests a general mono-decrease pattern from lower to higher latitudes, which explains the broader flattening of the isopycnal slope (Fig. 2j).

Similar to the SSP585 climate change, lighter surface density has expanded during the mPWP (Supplementary Fig. 5a and Fig. 4a). The lightening transformation, driven by surface heat and freshwater fluxes, extends further north than the future warming scenario (Supplementary Fig. 5b–d, and Fig. 4d). Consequently, water mass formation processes, including upwelling and downwelling, have also moved northward and are further north (Supplementary Fig. 5e, and Fig. 4e). The spatial compression of upwelling anomalies contributes to the slight weakening of the eddy-induced MOC response during the mPWP. Meanwhile, the downwelling anomalies around Antarctica have broadened, supporting more bottom water mass formation in this region than the SSP585 scenario (Supplementary Fig. 5f and Fig. 4f).

### Past and future changes in Southern Ocean ventilation
The Southern Ocean is one of the largest ventilation regions, where young surface waters are transported into the ocean interior. This

process is primarily driven by the overlying westerly winds. Ventilation can be quantified by ideal age, which represents the average time since the water last resided in the surface layer. Previous studies suggest that global warming increases the ideal age in polar waters due to enhanced stratification, which slows ventilation[34–36]. In contrast, the age of mode waters decreases, as confirmed by perturbation experiments involving intensified and shifted zonal wind stress[37,38].

Here, we investigate a subset of models with ideal age tracer output (Methods). We find that ideal age decreases along the subantarctic mode water path while increasing along the circumpolar deep water path from the historical to SSP585 scenario (Fig. 5), consistent with previous research[34]. This is likely due to the strengthened upwelling of water masses around Antarctica and thereby increased subduction in the subantarctic region. Compared to the future warming scenario, sub-duction is further enhanced, deeper, and occurring on higher latitudes during the mPWP (Fig. 5), with this effect being particularly pronounced in the IPSL-CM6A-LR simulation (Fig. 5c). Additionally, there is a decline in ideal age around the Antarctic continent, particularly in CESM1 models (Fig. 5a, b), indicating stronger downwelling and confirmed more bottom water formation during the mPWP compared to future warming scenario, as mentioned earlier. The difference highlights the complex interplay between wind forcings, buoyancy forcings, and ocean ventilation. In conclusion, either upwelling or subduction intensifies in the past and future warming scenarios, with a larger intensification during the mPWP than SSP585. The enhanced subduction suggests a potential increase in ocean heat and carbon uptake[36], which disrupts the carbon cycle and could act as feedback on global warming[39].

### Discussion
Our study highlights the intensification of the Southern Ocean residual MOC, with an eddy-induced MOC compensating for a poleward-strengthened Eulerian-mean MOC under future warming scenarios. This eddy compensation occurred in past warm periods, such as the mPWP, though the compensation pattern may differ from future climate. By analyzing buoyancy forcing, we find that the eddy-induced MOC is primarily modulated by surface heat flux change in lower latitudes but by freshwater flux change in higher latitudes in both past and future warmer climates. Corresponding to changes in surface buoyancy forcing, upwelling around the Antarctic shifts northward in both warmer climates, with an even greater displacement in the past, facilitating bottom water formation around the Ross and Weddell Seas. Additionally, ventilation via subantarctic mode water subduction has strengthened in both warmer climates, particularly during the mPWP.

Although the mPWP serves as a valuable natural analog to future anthropogenic warming, it is worth noting the difference between the two. For instance, polar ice sheets were substantially reduced during the mPWP, contributing to sea levels approximately 20 m higher than present[40]. This ice loss reflects a long-term quasi-equilibrium state under stable carbon dioxide levels. By contrast, SSP585 represents a transient warming scenario driven by rapidly rising greenhouse gas emissions[41]. A comparable reduction in the Antarctic icesheet is unlikely under the SSP585 due to the slow response time of ice sheets to forcing. The differing MOC response in the mPWP highlights the influence of ice around and from Antarctica on Southern Ocean circulation.

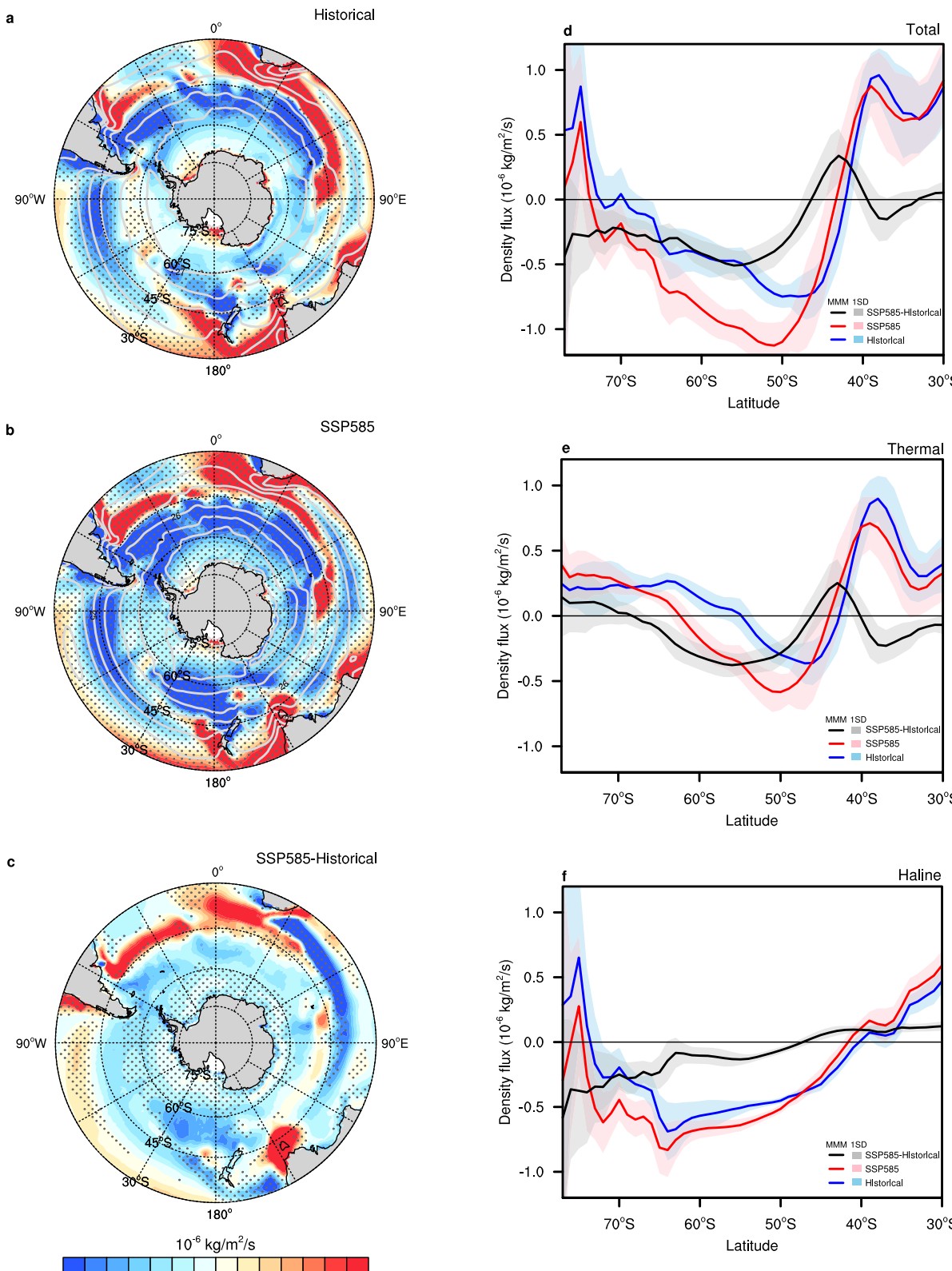

**Fig. 3 | Southern Ocean surface density flux from the historical to Shared Socioeconomic Pathway 5–8.5 (SSP585) scenario. a, b** Annual mean surface density fluxes (shaded, unit: 10⁻⁶ kg/m²/s) and surface potential density (gray contour, unit: kg/m³) for the multi-model mean (MMM) in CMIP5/6 **a** historical and **b** SSP585 simulations, as well as **c** the density flux difference between the two (SSP585 minus historical, shaded, unit: 10⁻⁶ kg/m²/s) over the Southern Ocean. **d** Annual and zonal mean surface density flux in historical (MMM, blue; inter-model spread, light blue) and SSP585 (MMM, red; inter-model spread, light red) simulations, and their difference (SSP585 minus historical, MMM, black; inter-model spread, gray). **e, f** Same as **d** but for the thermal and haline contributions to density flux, respectively. The stipples refer to the regions where at least two-thirds of the models agree with the sign of the MMM.

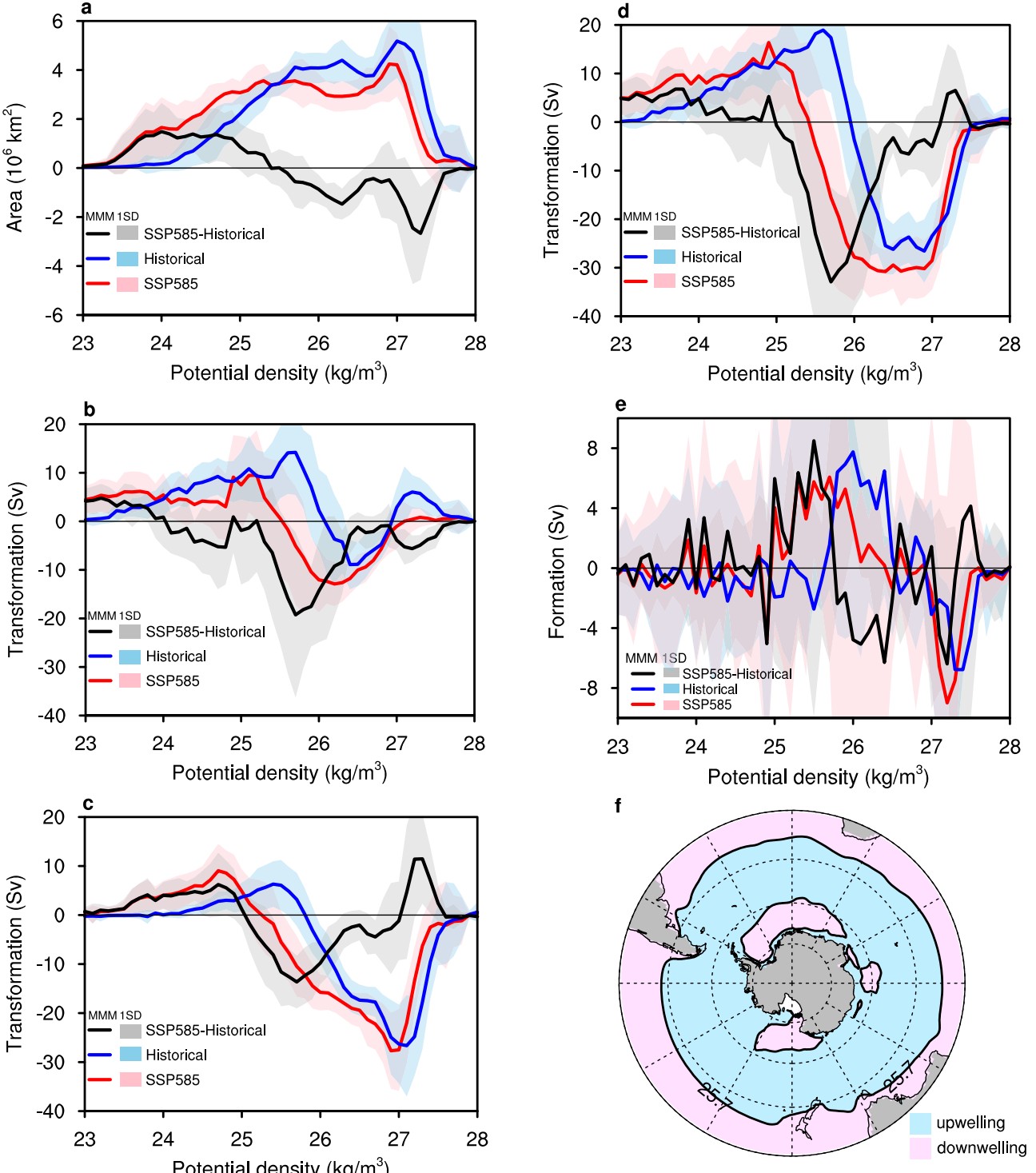

**Fig. 4 | Southern Ocean potential density area, water mass transformation, and formation from the historical to Shared Socioeconomic Pathway 5–8.5 (SSP585) scenario. a** Area of potential density in CMIP5/6 SSP585 (multi-model mean [MMM], red; inter-model spread, light red) and historical (MMM, blue; inter-model spread, light blue) simulations, and their difference (SSP585 minus historical, MMM, black; inter-model spread, gray) over the Southern Ocean. **b** Water mass transformation due to surface heat flux in SSP585 (MMM, red; inter-model spread, light red) and historical (MMM, blue; inter-model spread, light blue) simulations, and their difference (MMM, black; inter-model spread, gray). **c** Same as **b** but water mass transformation due to surface freshwater flux. **d** Same as **b** but the total water mass transformation with combined thermal and haline effects. **e** Water mass formation in SSP585 (MMM, red; inter-model spread, light red) and historical (MMM, blue; inter-model spread, light blue) simulations, and their difference (MMM, black; inter-model spread, gray). **f** Downwelling and upwelling regions, as indicated in (**e**).

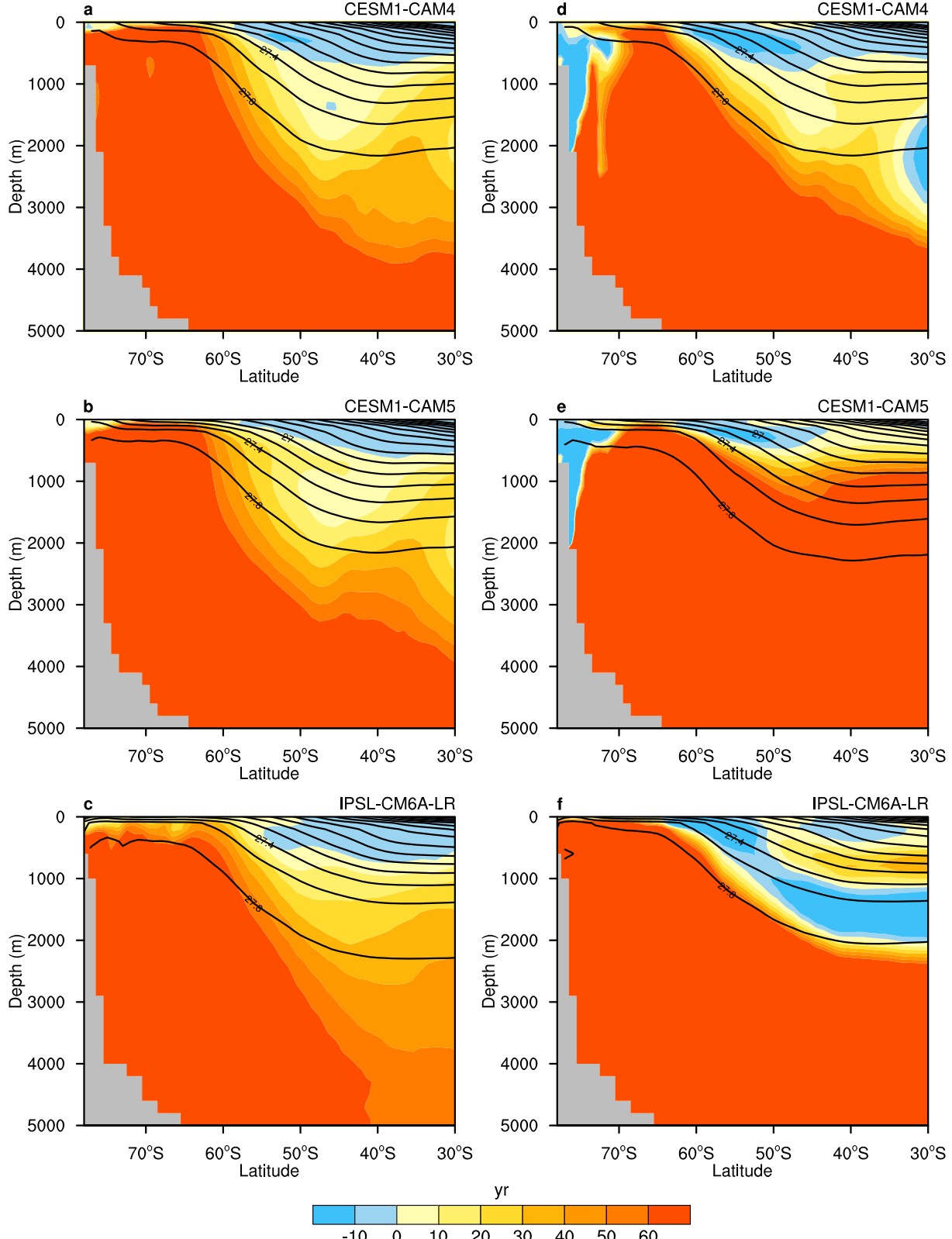

**Fig. 5 | Southern Ocean ideal age change under the Shared Socioeconomic Pathway 5–8.5 (SSP585) and the mid-Pliocene Warm Period (mPWP) scenarios.** **a**–**c** Zonal mean ideal age difference (shaded, units: yr) over the Southern Ocean between SSP585 and historical simulations (SSP585 minus historical) and historical potential density climatology (contour, units: kg/m³) with **a** CESM1–CAM4, **b** CESM1–CAM5 and **c** IPSL–CM6A–LR. **d**–**f** Same as **a**–**c** but for the difference between mPWP and preindustrial simulations (mPWP minus preindustrial) and preindustrial potential density climatology (contour, units: kg/m³).

Our findings indicate complex feedbacks that emerge from Southern Ocean MOC change. An intensified MOC can bring sequestered carbon from the deep ocean to the surface and further cause carbon dioxide outgassing[39,42,43], which, in turn, will modify atmospheric carbon dioxide concentration to change the climate and Southern Ocean MOC. Building on this, a focus should be placed on quantifying the sensitivity of air-sea carbon fluxes to changes in MOC strength and structure, particularly in regions of mode and intermediate water formation. Besides, the parameterization schemes used in current CMIP6 models may not accurately capture the full eddy response[44] while Antarctic icesheet dynamics have yet to be explicitly simulated in these models. Advances in model development will help refine the understanding of the Southern Ocean's role in future climate and carbon cycle trajectories.

## Methods

### ECCO reanalysis
We leverage the Estimating the Circulation and Climate of the Ocean (ECCO) version 4 release 4 (v4r4), a data-assimilative ocean state estimate that integrates observational data with a numerical ocean circulation model[45]. We examine the trends of the Eulerian-mean and eddy-induced MOCs, surface wind stress, and buoyancy flux over the Southern Ocean using monthly ECCO v4r4 output for the period 1992–2017.

### CMIP5/6 ensemble simulations
We analyze five suites of CMIP6 simulations: SSP585, SSP126, historical, mPWP experiments, as well as the preindustrial control run. The SSP585 represents a high-emissions scenario, which is regarded as a "business-as-usual" pathway with minimal climate mitigation efforts[46]. In contrast, the SSP126 represents a low-emissions scenario with climate mitigation efforts[46]. The mPWP experiment in CMIP6 focuses on simulating climate conditions approximately 3.2 million years ago, a time when Earth's average surface temperatures were estimated to be 1.8–3.6 °C warmer than preindustrial levels[13]. This period serves as an analog for understanding future climate under sustained high carbon dioxide levels.

We compare the average of 2081–2100 in SSP585/SSP126 with the average of 1981–2000 in the historical simulation, which allows us to assess future climate change in the context of observed and simulated recent historical climate evolution. Based on monthly outputs from SSP585/126 and historical simulations with 17 CMIP6 climate models (Supplementary Table 1), we calculate the ensemble mean for each model, and then compute the multi-model mean and inter-model spread. We also compare the last 100-year averages between mPWP and preindustrial simulations with 4 CMIP6 models (Supplementary Table 2). In addition to CMIP6 models, we include two CMIP5-era models, CESM1–CAM4 and CESM1–CAM5, in our study, since their mPWP simulation outputs contain an ideal age tracer (Supplementary Table 2). We exploit their RCP85 simulations but refer to SSP585 for the ease of discussion.

### Southern Ocean meridional overturning circulations
The Eulerian-mean MOC ($\bar{\psi}$) is calculated by integrating the Eulerian mean meridional velocity v zonal and vertically:

$$\bar{\psi}(y, z) = \oint \int_z^0 \bar{v} dz' dx \tag{1}$$

where $x, y, z'$ are the zonal, meridional, and vertical coordinates, and $z$ denotes an arbitrary depth.

Similarly, the eddy-induced MOC ($\psi^*$) is calculated as:

$$\psi^*(y, z) = \oint \int_z^0 v^* dz' dx \tag{2}$$

where $v^*$ eddy-induced velocity. The residual MOC ($\psi_{res}$) is the sum of Eulerian-mean and eddy-induced MOCs:

$$\psi_{res} = \bar{\psi} + \psi^* \tag{3}$$

For CMIP5/6 models used in the current study, mesoscale oceanic eddies are parameterized by the Gent–McWilliams (GM) scheme, and eddy-induced velocity is represented by bolus velocity, which varies spatially and temporally and is a function of the slope of the local isentropic surface[47]. This eddy effect on ocean circulation has been included in the CMIP6 monthly output of meridional mass streamfunction. As a result, the residual MOC ($\psi_{res}$) is directly derived from the model output of meridional mass streamfunction, the Eulerian-mean MOC ($\bar{\psi}$) is computed directly from the monthly-mean meridional velocity field (Eq. (1)), and the eddy-induced MOC ($\psi^*$) is then obtained as the difference between the residual and Eulerian-mean MOCs (Eq. (3)). Note here, the Eulerian-mean MOC refers to the time-mean circulation, and the eddy-induced MOC refers to the time-varying circulation.

### The slope of the mean buoyancy surface
According to ref. 5, the eddy-induced MOC ($\psi^*$) can be written as:

$$\psi^* = -\frac{\overline{w'b'}}{\overline{b_y}} = \frac{\overline{v'b'}}{\overline{b_z}} = -K\frac{\overline{b_y}}{\overline{b_z}} = K s_\rho \tag{4}$$

where $\overline{w'b'}$ and $\overline{v'b'}$ are vertical and meridional eddy buoyancy fluxes, respectively, $\overline{b_y}$ and $\overline{b_z}$ are mean meridional and vertical buoyancy gradients, respectively, $K$ represents an eddy transfer coefficient, and the slope of the mean buoyancy surface ($s_\rho$) can be calculated by:

$$s_\rho = -\frac{\overline{b_y}}{\overline{b_z}} \tag{5}$$

Here, the buoyancy ($b$) is defined by:

$$b = -g\frac{\rho}{\rho_0} \tag{6}$$

where g denotes gravity acceleration, $\rho$ is the potential density of ocean water referenced to the surface, and $\rho_0$ is freshwater density with a value of 1000 kg m$^{-3}$.

### Surface density flux
Surface density flux quantifies the gain or loss of water mass in the ocean's surface layer resulting from heat and freshwater exchanges[22]:

$$D = -\frac{\alpha}{c_p}H + SSS\frac{\rho_s}{\rho_0}\beta FW \tag{7}$$

where $\alpha$ and $\beta$ denote the thermal expansion and haline contraction coefficients, respectively, and $c_p$ denotes the specific heat capacity. SSS is sea surface salinity, and $\rho_s$ is the surface density of ocean water.

The term $H$ accounts for surface heat flux components: shortwave and longwave radiation, latent and sensible heat, and residual (Supplementary Fig. 1), wherein the residual term includes heat fluxes due to frazil ice formation/melt. The term FW encompasses surface freshwater fluxes: evaporation, precipitation, runoff, snowfall, and residual (Supplementary Fig. 2), wherein the runoff term includes river runoff as well as meltwater from icebergs and ice shelves, and the residual term includes sea ice melt and brine rejection. A positive density flux indicates a lightening of ocean water due to warming or freshening, while a negative density flux signifies densification caused by cooling or salinification.

## Water mass transformation and formation

According to the previous study[22], water mass transformation is defined as the volume flux of a water mass that is either consumed or produced by buoyancy forcing within a specific density class. Water mass formation refers to the convergence or divergence of these transformed water masses, which can drive subduction or upwelling through the base of the ocean's mixed layer.

The averaged water mass transformation rate $F$ in a given potential density class ($\sigma_0$) is calculated by:

$$F(\sigma_0) = \iint dxdyD\,(x,y)\delta(\sigma(x,y) - \sigma_0) \tag{8}$$

where $\delta$ is a delta function equal to zero except when potential density ($\sigma$) is within the range $[\sigma_0 - \frac{1}{2}\Delta\sigma: \sigma_0 + \frac{1}{2}\Delta\sigma]$.

The averaged water mass formation $M(\sigma)$ is defined as the water that accumulate between successive isopycnals $\sigma_1$ and $\sigma_2$:

$$M(\sigma) = -[F(\sigma_2) - F(\sigma_1)] \tag{9}$$

## Significance test

To assess the robustness of the diagnosed trends, we apply a two-tailed Student's t-test to evaluate statistical significance at the 90% confidence level. In addition, inter-model agreement is considered significant where at least two-thirds of the models concur on the sign of the multi-model mean response.

## Data availability

All the raw data used in this study are publicly available online. The CMIP5 model outputs for RCP8.5, historical, preindustrial, and mPWP simulations used in this study are available in the Earth System Grid Federation (ESGF) database under accession https://esgf-node.llnl.gov/projects/cmip5/. The CMIP6 model outputs for SSP585, SSP126, and historical simulations used in this study are available in the ESGF database under accession https://esgf-node.llnl.gov/projects/cmip6/. The ECCO v4r4 ocean state estimate used in this study is available in the NASA Physical Oceanography Distributed Active Archive Center (PODAAC) database under accession https://podaac.jpl.nasa.gov/ECCO/. The processed datasets used to generate all figures in this study (Source Data) are available from Zenodo[48] at https://doi.org/10.5281/zenodo.17246188.

## Code availability

The code used in this study is available from https://doi.org/10.5281/zenodo.17246188.

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

## Acknowledgements
This research is supported by the US National Science Foundation (NSF OCE-2123422, AGS-2053121, and AGS-2237743) awarded to W.L. The authors are grateful to all data providers.

## Author contributions
W.L. conceived the initial idea. T.T.Z. performed the analysis and wrote the manuscript. Both authors contributed to the improvement of the paper.

## Competing interests
The authors declare no competing interests.
