## [Transparent Peer Review file · Nature Communications]

Evolving Southern Ocean overturning in warming climates

Corresponding Author: Dr Tingting Zhu

Version 0:

Reviewer comments:

Reviewer #1

(Remarks to the Author)

Review of Evolving Southern Ocean overturning in warming climates, by Tingting Zhu, Wei Liu

This manuscript first quantifies the MOC in the Southern Ocean and decomposes it into eddy and mean components. The effects of warming are assessed in two scenarios: anthropogenic climate change under SSP585 of CMIP6 and the mid-Pliocene warm period. The drivers of the changes in MOC are investigated by calculating changes in surface density flux and water mass transformation. These values are further broken down into the contribution of thermal and haline components, as well as latent and sensible heat flux and longwave and shortwave radiation. Finally, the ideal age of water masses throughout the Southern Ocean is plotted to identify changes in ventilation.

The paper is relatively clearly written, although the results section should be reviewed for concision. These results are potentially significant for a variety of reasons including carbon sequestration, heat transport, and regional climate change. I expect that other researchers in the field would find the results interesting. The methods appear relatively simple, but are not explained in detail in the manuscript, which prevents me from fully evaluating them. In particular, the assessment of the eddy-component of any process is perplexing considering the data comes exclusively from models which do not resolve eddy activity. For this reason primarily, I find that this manuscript needs major revisions before it can be considered for publication. Detailed comments are below.

Major comments

- The largest point of uncertainty in this analysis is that the authors attempt to decompose the mean- and eddy-components of MOC using datasets that are not eddy-resolving. I see in one previous work from the authors that the eddy contribution to velocity is derived from the GM eddy parameterization, but that is not made clear in this manuscript. This will be crucial to address before publication can be considered. This weakness is compounded by the vague methods.
- The methods are insufficient, in my opinion, to reproduce this work. The specific variables used are not given, which confuses how mean and eddy-components are defined. What is mentioned, is that monthly data is used, which implies that the “eddy-contribution” to MOC is derived from data that is both spatially and temporally insufficient to resolve eddy activity. The code has also not been made public, meaning I cannot investigate the methods more directly than what is written in the text.
- Throughout the manuscript, the authors make the distinction between Eulerian and eddy components. Taking Eulerian to mean the quantification of the properties of a water body at fixed points, as opposed to Lagrangian, the characterization of distinct water parcels that may move in space over time, I don't think the language used in the manuscript is accurate. Eddy activity can be quantified using Eulerian methods. In fact, it seems that the authors use the Eulerian decomposition of time-mean and time-varying conditions to identify the eddy-contribution to MOC. The authors also use the term Eulerian-mean quite often, but I will point out that this does not necessarily imply a time-mean - it could also imply a spatial mean over fixed grid points. I suggest explaining that the mean and eddy-components are defined as “time-mean” and “time-varying”, if that is indeed the case.
- To this point, it is ambiguous in the manuscript how velocity is decomposed, as the term “eddy-induced velocity” is used at line 315. These methods must be made clear.
- The analysis of SSP126 adds very little information to the manuscript. The pattern is similar to the SSP585 with lower magnitudes. Repeating the information in the text seems unnecessary.
- I suggest that the authors review the results section and try to reduce repetition.
- One of the major aspects of the study is a comparison between anthropogenic warming and the mid-pliocene warm period. These two climatic states produce relatively distinct changes from their reference datasets, but very little explanation for

these differences is included in the text. I don't see the value of including both of these datasets in the study without trying to determine what the substantial differences are that create the discrepancy, or to look for similarities that may shed light on the future climate under anthropogenic warming.

- Given that the mPWP simulations are compared to a preindustrial climate, would it not be more appropriate to compare SSP585 simulations to a pre-industrial control from CMIP?
- It is stated that changes in heat flux are primarily responsible for density changes and the resulting changes in upwelling and downwelling regions. Yet, in both simulations analyzed, freshwater flux appears to be the dominant driver around the Antarctic continent (Figure 3, supplementary Figure 5). This is unsurprising given the impact of warming on meltwater from Antarctic ice. I suggest modifying the text to acknowledge this.
- Figure 1a. The different axes on this plot make it somewhat confusing without a close look. I suggest plotting the trend and mean on the same axis to give an impression of the scale of change relative to present conditions.
- Figure 1b,c,d. The contour lines here do not show the historical magnitude of MOC. I find this quite important for interpreting changes.
- Only Figure 1 includes a measure of statistical significance in the figures. I suggest adding something similar to all of the plots.

Minor comments

Line 65. "weakened" should be "weaken"

Figure 1. In the caption, e and f should be f and g.

Line 108. Is diminished a mistake. It seems to me that diminished MOC does not compensate for weakened MOC.

Line 110. How is it determined what changes are negligible?

Line 112. (Methods) doesn't seem like an adequate explanation for this claim to me.

Lines 150-160. The uncertainty in figure 4e is very large. I would question whether the results in this paragraph are reliable. At the least, the large disagreement in the model ensemble should be addressed in the text.

Line 244. This is the only place where individual models are mentioned. I am not sure what value this statement adds to the text.

Line 272. Caused should be cause.

Line 339. I don't think freshening is the correct word, since heat may also be responsible for these changes.

Supplementary Figure 4. I expect that runoff in this figure is not from rivers.

Most figures use mPWP and PI as abbreviations and are referred to before these terms are defined in the text. It would be helpful if the figures could stand on their own without needing extra context from later in the manuscript.

Figure 3. The red contours on panels a and b are not explained.

Figure 4. b-d are described as "same as a" but have different units on the y axis. I suggest describing the panels b-d specifically and individually.

(Remarks on code availability)

This doi does not work for me.

Reviewer #2

(Remarks to the Author)

This is my first review of the manuscript by Zhu and Liu, titled 'Evolving Southern Ocean overturning in warming climates'. In this work the authors use a subset of CMIP6 models to study the sensitivity of Southern Ocean Meridional Overturning Circulation (MOC) to increasing greenhouse gas emission in past and future warm climates. The authors state that the main finding of their work is the poleward shifting Eulerian MOC that is partly compensated by the eddy driven MOC (less so in low emission or in a past warm climate). As a consequence the authors argue that also Southern Ocean ventilation intensifies. I find the manuscript interesting and in principle well written. To me the arguments towards changing ventilation are the most interesting and novel as they could have global consequences. However, as it is now, I don't think the ventilation analysis is robust – I have made a suggestion to improve that (see below). In addition, it remains unclear how the authors treat sea ice in their analysis, their reference list is missing a few review papers (such as <https://doi.org/10.1098/rsta.2013.0296>), and could do a better job in commenting on the fact that the CMIP models do not resolve eddies (<https://journals.ametsoc.org/view/journals/phoc/40/7/2010jpo4353.1.xml>). Other than that, I have several minor comments that should be addressed, not least the code statement. I believe after these revisions, the manuscript could be considered for publication. Please find my detailed comments below.

MAJOR

L239, Fig. 5, Supplementary Fig. 9. Ideal age is a very useful tracer, but its interpretation can be a bit difficult. The absolute age integrates the history of the simulation, and may not be so useful. For example, it remains unclear if Fig5/Supplementary Fig 9 plot the difference between hist and SSP585 climatologies, or values at the end of both simulations. I think it is the former, in which case the plot shows that oldest water masses are aging most – this is somewhat trivial and percentage difference would probably be a more meaningful measure. The problem with the absolute values is also to know if they are initialized to zero at the same time. The historical simulation is not 400 years old, yet the climatology shows 400 year contour. It also remains unclear if the negative values are a real signal of changing ventilation or just a shift in the age gradient.

To resolve the issues linked to the absolute age, I would suggest the authors to use the trend (time derivative) of the ideal age instead. This trend gives the rate at which a watermass is aging at any given time. This would allow the authors to

compare the SSP585 and mPWP ventilation to PI ventilation directly. The authors could compute this point-wise in their zonal section, but perhaps stating a global change in ventilation would be meaningful as the authors elude to possibly increasing ocean carbon/heat uptake (which depends on other regions than SO as well).

L133 and L337 The authors seem to ignore sea ice melt (or is that the residual? Please add some supplementary text to explain the supplementary figures). However, I would expect that sea ice melt should be a rather large source of fresh water in SSP585 and to affect the surface density flux – Fig. 3 d-f seems to suggest that south of 65S the freshwater dominates the budget and I would expect that sea ice melt, rather than P-E would be dominating.

MINOR

In fig 1 I would suggest adding 'negative contours are dashed' and then in the text I would mention that Eulerian MOC is clockwise (positive) in the mean and the eddy MOC is anticlockwise (negative) in the mean. This would make it easier for the reader to digest fig 1 and realize that both positive on positive contours, and negative on negative contours, indicate strengthening. Also, I would suggest mentioning that the robust trend is the intensification of the residual MOC between 40-60S due to the intensification of the Eulerian MOC. The eddy MOC trends are hardly significant.

L104 the reader is referred to Fig 2 here, but once arriving to Fig 2 the reader is left wondering what mPWP is until reading the text to L172. Since the term mPWP is quite prominent in Fig 2, I would suggest helping the reader a bit and spelling out the acronym in the caption to avoid unnecessary confusion.

L86 (and thereafter) although ECCO is probably our best estimate of the SO state, I wouldn't called it 'observations', rather 'observationally constrained estimate of the Eulerian MOC...' or something similar.

L178 'a little bit' is not very precise, please rephrase.

L272 should probably read 'cause' instead of 'caused'

L65 'weakened' should read 'weaken'

L267-L278 brings very little added value to the manuscript. I would suggest that the authors try to tie their work to existing literature a bit tighter and give clear recommendations for future work, not just list very broad topics as is done now.

L338 I would ask the authors to be exact when listing what H includes, it seems to include a few things 'such as...', but the reader is left wondering if this is the complete list or if other terms contribute (and it is also unclear whether or not only the listed things are included or if something else is also included). It is also not clear what residual is in this case.

L359-L361 The location of mPWP data is not given. I would ask the authors to state how this data can be downloaded.

L364-365 I have to say that the code availability statement sounds odd. I would strongly recommend the authors make their code available through an online repository, such as Zenodo, with a citable DOI.

(Remarks on code availability)

The code was not available and I have commented on it in my review

Version 1:

Reviewer comments:

Reviewer #1

(Remarks to the Author)

The authors have made a respectable effort to address the issues raised in the previous round of review and have improved their manuscript substantially. I am generally satisfied with the revisions and can recommend that the manuscript be published.

(Remarks on code availability)

Point-by-Point Replies to Reviewer #1

This manuscript first quantifies the MOC in the Southern Ocean and decomposes it into eddy and mean components. The effects of warming are assessed in two scenarios: anthropogenic climate change under SSP585 of CMIP6 and the mid-Pliocene warm period. The drivers of the changes in MOC are investigated by calculating changes in surface density flux and water mass transformation. These values are further broken down into the contribution of thermal and haline components, as well as latent and sensible heat flux and longwave and shortwave radiation. Finally, the ideal age of water masses throughout the Southern Ocean is plotted to identify changes in ventilation.

The paper is relatively clearly written, although the results section should be reviewed for concision. These results are potentially significant for a variety of reasons including carbon sequestration, heat transport, and regional climate change. I expect that other researchers in the field would find the results interesting. The methods appear relatively simple, but are not explained in detail in the manuscript, which prevents me from fully evaluating them. In particular, the assessment of the eddy-component of any process is perplexing considering the data comes exclusively from models which do not resolve eddy activity. For this reason primarily, I find that this manuscript needs major revisions before it can be considered for publication. Detailed comments are below.

Reply: We sincerely appreciate the reviewer's thoughtful input, which has led to substantial improvements in the manuscript. In response to the reviewer's feedback, we have clarified in the Methods section how each component of the MOC is calculated, clarified eddy parameterizations in CMIP6 models, and updated the available code to enhance reproducibility. We believe these revisions have strengthened the manuscript and effectively addressed the reviewer's concerns.

Major comments:

Comment: (1) The largest point of uncertainty in this analysis is that the authors attempt to decompose the mean- and eddy-components of MOC using datasets that are not eddy-resolving. I see in one previous work from the authors that the eddy contribution to velocity is derived from the GM eddy parameterization, but that is not made clear in this manuscript. This will be crucial to address before publication can be considered. This weakness is compounded by the vague methods.

Reply: We agree with the reviewer that the models used in our study are not eddy-resolved. The Eulerian-mean overturning is computed directly from the monthly-mean ocean meridional velocity field (v_o). The residual-mean overturning is taken from the model output of ocean meridional/y overturning mass streamfunction ($msftmz/msfty_z$), which includes the eddy-induced transport as parameterized by the Gent–McWilliams (GM) scheme in CMIP6 models. The eddy-induced MOC is then obtained as the difference between residual and Eulerian components. We have updated the Methods section to clarify this methodology and explicitly state that the eddy contribution is derived from the GM eddy parameterization (Lines 321-330 of the revised manuscript).

Comment: (2) The methods are insufficient, in my opinion, to reproduce this work. The specific variables used are not given, which confuscates how mean and eddy-components are defined. What is mentioned, is that monthly data is used, which implies that the “eddy-contribution” to MOC is derived from data that is both spatially and temporally insufficient to resolve eddy activity. The code has also not been made public, meaning I cannot investigate the methods more directly than what is written in the text.

Reply: Following the review, we have carefully revised the manuscript to address these concerns in the following ways. The Eulerian-mean MOC is computed using the monthly-mean meridional velocity field. The residual-mean MOC is taken directly from the CMIP5/6 diagnostic variable (meridional mass streamfunction), which includes the eddy-induced transport derived from the Gent–McWilliams (GM) eddy parameterization. The eddy-induced overturning streamfunction is computed as the difference between the residual MOC and the Eulerian-mean MOC. In models, transient eddies are represented by bolus velocities from GM eddy parameterization, which vary spatially and temporally during diagnostics and time step iterations. Such eddy part has been included in the monthly output of meridional mass streamfunction. By calculating the difference between the residual and Eulerian MOCs, we are able to extract the MOC caused by transient eddies. We have updated the Methods section to clarify this methodology (Lines 321-330 of the revised manuscript).

Besides, to enhance reproducibility, we have provided the analysis code in a publicly accessible repository, as noted in the Code Availability section. It will be publicly available upon publication. The reviewer can access it during review via the provided link:
<https://zenodo.org/records/16734876?token=eyJhbGciOiJIUzUxMiJ9.eyJpZCI6IjQ3MDNiOTJmLWE1ZjQtNGY5Mi05OTI3LWM5OTM4YTc0MDgwYSIsImRhdGEiOnt9LCJyYW5kb20iOiI3ZjFjMWU1Y2FmZTAzMzNjZWViNzQ2N2I4OWU2OTI1MSJ9.hx1EylFfqXHAqKQ680OXoj4JlFxBnKB7hAIyUfQktalqDsX3ZKIn-K69z-TQIFuAu5jhdB0NMfvWKdTuvQ9rzQ>

Comment: (3) Throughout the manuscript, the authors make the distinction between Eulerian and eddy components. Taking Eulerian to mean the quantification of the properties of a water body at fixed points, as opposed to Lagrangian, the characterization of distinct water parcels that may move in space over time, I don't think the language used in the manuscript is accurate. Eddy activity can be quantified using Eulerian methods. In fact, it seems that the authors use the Eulerian decomposition of time-mean and time-varying conditions to identify the eddy-contribution to MOC. The authors also use the term Eulerian-mean quite often, but I will point out that this does not necessarily imply a time-mean - it could also imply a spatial mean over fixed grid points. I suggest explaining that the mean and eddy-components are defined as “time-mean” and “time-varying”, if that is indeed the case.

Reply: We agree with the reviewer and thus clarified in the revision that "Eulerian-mean" also includes spatial averaging over fixed grid points. In our manuscript, the term "Eulerian-mean" refer specifically to the time-mean meridional overturning streamfunction computed from the monthly-mean velocity field at fixed spatial locations. We have revised the Method section to explicitly state that the Eulerian-mean MOC refers to the time-mean circulation and the eddy-induced MOC refers to the time-varying circulation (Lines 328-330 of the revised manuscript).

Comment: (4) To this point, it is ambiguous in the manuscript how velocity is decomposed, as the term “eddy-induced velocity” is used at line 315. These methods must be made clear.

Reply: As mentioned in our reply for Comment #3, the eddy-induced component is inferred as the difference between the residual-mean MOC and the time-mean Eulerian MOC, which manifests the GM parameterized bolus velocity. We have clarified this in the Methods section.

Comment: (5) The analysis of SSP126 adds very little information to the manuscript. The pattern is similar to the SSP585 with lower magnitudes. Repeating the information in the text seems unnecessary.

• I suggest that the authors review the results section and try to reduce repetition.

Reply: Agree. Following the review, we have revised the results section related to the SSP126 to reduce repetition (Lines 165-172 of the revised manuscript).

• One of the major aspects of the study is a comparison between anthropogenic warming and the mid-pliocene warm period. These two climatic states produce relatively distinct changes from their reference datasets, but very little explanation for these differences is included in the text. I don't see the value of including both of these datasets in the study without trying to determine what the substantial differences are that create the discrepancy, or to look for similarities that may shed light on the future climate under anthropogenic warming.

Reply: The motivation for including both the mPWP and the SSP585 scenario is to compare the Southern Ocean MOC responses under two distinct warm climate states. The comparison allows us to assess the extent to which ocean circulation features such as Southern Ocean overturning, bottom water formation, and ventilation are consistent or divergent under warm conditions with different forcing mechanisms. As the reviewer notes, the differences between these two states are substantial, and we agree that further discussion is warranted. To address the reviewer's concern, we have added a discussion of key similarities and differences between the past and future warming climates in both result and discussion sections.

• Given that the mPWP simulations are compared to a preindustrial climate, would it not be more appropriate to compare SSP585 simulations to a pre-industrial control from CMIP?

Reply: In this study, the mPWP simulations are compared to the pre-industrial (PI) control run, while the SSP585 simulations are compared to the historical run. This comparison framework follows the design logic of the CMIP6 experimental protocols and the nature of each experiment. The mPWP simulations represent a quasi-equilibrium state of Earth's climate under natural forcings, without any anthropogenic influence. Therefore, the preindustrial control, which also excludes anthropogenic forcing and represents a pre-industrial steady-state climate, serves as the appropriate baseline for isolating structural and boundary-forced climate differences between the two time periods.

In contrast, the SSP585 simulation is a transient, forced projection initialized directly from the end of the historical simulation, which incorporates anthropogenic forcings (greenhouse gases, aerosols, etc.) and observed variability. Thus, comparing the SSP585 to historical simulation, rather than the preindustrial control, is better suited to assess future anthropogenic impacts

relative to the recent climate baseline, which is more relevant for understanding near-term changes and societal impacts.

We understand the reviewer’s point about comparing the SSP585 to the preindustrial simulation so that both analyses will use the same control. Here, we present the responses of the Southern Ocean MOC and wind stress in the SSP585 relative to preindustrial control (Fig. A1), which are consistent with the responses relative to the historical simulation (Fig. 2 in the revision), albeit with larger magnitude. This result strengthens the robustness of our conclusion.

Fig. A1 | **a** Annual and zonal mean surface zonal wind stress in CMIP5/6 SSP585 (multi-model mean or MMM, red; inter-model spread, light red), preindustrial (MMM, blue; inter-model spread, light blue) simulations, and their difference (SSP585 minus preindustrial, MMM, black; inter-model spread, gray) over the Southern Ocean. The inter-model spread is defined as one standard deviation among models. **b** Difference in annual mean Southern Ocean Eulerian-mean MOC (shaded, units: Sv) for the MMMs between SSP585 and preindustrial simulations, overlaid with the annual mean preindustrial Eulerian-mean MOC climatology (contours, units: Sv). **c** same as **b**, but for the eddy-induced MOC. **d** same as **b**, but for the residual MOC. The stipples refer to the regions where at least two thirds of the models agree with the sign of the difference in the multi-model mean.

- It is stated that changes in heat flux are primarily responsible for density changes and the resulting changes in upwelling and downwelling regions. Yet, in both simulations analyzed, freshwater flux appears to be the dominant driver around the Antarctic continent (Figure 3,

supplementary Figure 5). This is unsurprising given the impact of warming on meltwater from Antarctic ice. I suggest modifying the text to acknowledge this.

Reply: We agree that freshwater flux plays a dominant role in density changes around the Antarctic margin due to melts from sea ice, icebergs, and ice shelves in a warming climate. We also make a note that surface freshwater flux change around Antarctica in CMIP6 models may be subject to uncertainty, as most models do not explicitly simulate Antarctic icesheet melting processes (Lines 139-141 of the revised manuscript).

Comment: (6) Figure 1a. The different axes on this plot make it somewhat confusing without a close look. I suggest plotting the trend and mean on the same axis to give an impression of the scale of change relative to present conditions.

Reply: We thank the reviewer's suggestion. Since the magnitude of the trend and mean have a large difference and also have different units, we used different axes to reflect not only their values but also the shifting trend in one figure. Using the same axis would make the shifting trend less discernible. To address the reviewer's concern, we have emphasized the different axis for the trend and mean in the Figure 1 caption.

Comment: (7) Figure 1b,c,d. The contour lines here do not show the historical magnitude of MOC. I find this quite important for interpreting changes.

Reply: Following the review, we have added contour labels to indicate the values in the Figures 1b-d.

Comment: (8) Only Figure 1 includes a measure of statistical significance in the figures. I suggest adding something similar to all of the plots.

Reply: Following the review, we have added significance tests to all figures. In the figures, stippling indicates regions where at least two-thirds of the models agree on the sign of the multi-model mean or the change.

Minor comments:

Comment: (1) Line 65. "weakened" should be "weaken"

Reply: Revised as suggested.

Comment: (2) Figure 1. In the caption, e and f should be f and g.

Reply: Revised as suggested.

Comment: (3) Line 108. Is diminished a mistake. It seems to me that diminished MOC does not compensate for weakened MOC.

Reply: We understand the reviewer's concern, while "diminished" here is not a mistake in this context. The eddy-induced MOC represents a counter-clockwise circulation, which opposes the clockwise circulation of the Eulerian-mean MOC. When the eddy-induced MOC is weakened (i.e., shows positive anomalies), it indicates a weaker counter-clockwise circulation, effectively resulting in a net clockwise anomaly. This can partially compensate for the weakened Eulerian-

mean MOC (weakened clockwise circulation). We have revised the text and included a more detailed description of the MOC (Lines 58-61 and 118-120 of the revised manuscript).

Comment: (4) Line 110. How is it determined what changes are negligible?

Reply: Near 60°S, the change in the eddy-induced MOC is much smaller in magnitude than that in the Eulerian MOC, especially in the upper 3000 m. We understand the wording “negligible” might be confusing, and thus rephrase the sentence in the text (Lines 123-125 of the revised manuscript): “Nonetheless, the enhanced eddy-induced MOC near 60°S offers a smaller compensation (Fig. 2d), thus the residual MOC predominantly alters with the Eulerian-mean MOC.”.

Comment: (5) Line 112. (Methods) doesn’t seem like an adequate explanation for this claim to me.

Reply: According to the review, we add the explanation in the Lines 120-123 of the revised manuscript: “This is especially pertinent to the varying surface buoyancy forcing, which combines changes in surface freshwater and heat fluxes, modulates Southern Ocean density structure and baroclinicity, and, in particular, reduces the isopycnal slope around 40°S (Fig. 1d), slowing the eddy-induced MOC there (Method).”.

Comment: (6) Lines 150-160. The uncertainty in figure 4e is very large. I would question whether the results in this paragraph are reliable. At the least, the large disagreement in the model ensemble should be addressed in the text.

Reply: We acknowledge that Figure 4e shows substantial inter-model spread. Following the review, we have revised the text to explicitly mention the uncertainty and the ensemble spread: “Besides, subduction anomalies appear around Antarctica, particularly in the Ross Sea and Weddell Sea, which are prompted by a northward displacement of upwelling and thereby support the bottom water mass formation in these regions (Fig. 4e and 4f), despite the fact that uncertainty may remain regarding how different models simulate water mass formation.”

Comment: (7) Line 244. This is the only place where individual models are mentioned. I am not sure what value this statement adds to the text.

Reply: Since only a few models—IPSL-CM6A-LR, CESM1-CAM4, and CESM1-CAM5—have simulated the ideal age tracer for the mPWP, it is necessary to assess the performance of each model individually to determine whether they show consistent behavior or if any of them is an outlier. We have clarified this in the text “Here, we investigate a subset of models with ideal age tracer output (Methods)” and also revised Fig. 5 to explicitly show each model’s result.

Comment: (8) Line 272. Caused should be cause.

Reply: Revised as suggested.

Comment: (9) Line 339. I don’t think freshening is the correct word, since heat may also be responsible for these changes.

Reply: Following the review, we have changed “freshening” to “lightening” to reflect the density flux change due to the heat and freshwater flux.

Comment: (10) Supplementary Figure 4. I expect that runoff in this figure is not from rivers.

Reply: Yes, the runoff term includes river runoff as well as meltwater from icebergs and ice shelves. We have clarified this in the Method section as well as figure caption.

Comment: (11) Most figures use mPWP and PI as abbreviations and are referred to before these terms are defined in the text. It would be helpful if the figures could stand on their own without needing extra context from later in the manuscript.

Reply: Revised as suggested.

Comment: (12) Figure 3. The red contours on panels a and b are not explained.

Reply: The red contours are annual mean surface potential density (unit: kg/m^3) for the MMMs in the historical (Fig. 3a) and SSP585 (Fig. 3b). In response to the review, we have added a description of the red contours in Figure 3's caption.

Comment: (13) Figure 4. b-d are described as "same as a" but have different units on the y axis. I suggest describing the panels b-d specifically and individually.

Reply: Revised as suggested.

Comment: (14) Reviewer #1 (Remarks on code availability): This doi does not work for me.

Reply: Yes, we have updated a valid DOI in the revision. It will be publicly available upon publication. The reviewer can access it during review via the provided link:

<https://zenodo.org/records/16734876?token=eyJhbGciOiJIUzUxMiJ9.eyJpZCI6IjQ3MDNiOTJmLWE1ZjQtNGY5Mi05OTI3LWw5OTM4YTc0MDgwYSIsImRhdGEiOiI9LjYyY5kb20iOiI3ZjFjMWU1Y2FmZTAzMzNjZWVhbnZQ2N2I4OWU2OTI1MSJ9.hx1EylFfqXHAqKQ680OXoj4J1FxBnKB7hAIyUfQktalqDsX3ZKIn-K69z-TQIFuAu5jhdB0NMfvWKdTuvQ9rzQ>

Point-by-Point Replies to Reviewer #2

This is my first review of the manuscript by Zhu and Liu, titled ‘Evolving Southern Ocean overturning in warming climates’. In this work the authors use a subset of CMIP6 models to study the sensitivity of Southern Ocean Meridional Overturning Circulation (MOC) to increasing greenhouse gas emission in past and future warm climates. The authors state that the main finding of their work is the poleward shifting Eulerian MOC that is partly compensated by the eddy driven MOC (less so in low emission or in a past warm climate). As a consequence the authors argue that also Southern Ocean ventilation intensifies. I find the manuscript interesting and in principle well written. To me the arguments towards changing ventilation are the most interesting and novel as they could have global consequences. However, as it is now, I don’t think the ventilation analysis is robust – I have made a suggestion to improve that (see below). In addition, it remains unclear how the authors treat sea ice in their analysis, their reference list is missing a few review papers (such as <https://doi.org/10.1098/rsta.2013.0296>), and could do a better job in commenting on the fact that the CMIP models do not resolve eddies (<https://journals.ametsoc.org/view/journals/phoc/40/7/2010jpo4353.1.xml>). Other than that, I have several minor comments that should be addressed, not least the code statement. I believe after these revisions, the manuscript could be considered for publication. Please find my detailed comments below.

Reply: We sincerely appreciate the reviewer’s insightful comments and suggestions, which have helped us improve the quality of the manuscript. In response to the reviewer’s feedback, we have enhanced the analysis of the ideal age tracer. We have also added a discussion on the role of sea ice melting. Additionally, we have clarified in the Methods section that the CMIP6 models used in our study are not eddy-resolved. We have also added the references the reviewer mentioned. We believe these revisions have strengthened the manuscript and addressed the reviewer’s concerns.

Major comments:

Comment: (1) L239, Fig. 5, Supplementary Fig. 9. Ideal age is a very useful tracer, but its interpretation can be a bit difficult. The absolute age integrates the history of the simulation, and may not be so useful. For example, it remains unclear if Fig5/Supplementary Fig 9 plot the difference between hist and SSP585 climatologies, or values at the end of both simulations. I think it is the former, in which case the plot shows that oldest water masses are aging most – this is somewhat trivial and percentage difference would probably be a more meaningful measure. The problem with the absolute values is also to know if they are initialized to zero at the same time. The historical simulation is not 400 years old, yet the climatology shows 400 year contour. It also remains unclear if the negative values are a real signal of changing ventilation or just a shift in the age gradient.

Reply: We agree with the reviewer and understand the reviewer’s concern about the absolute age. Ideal age is a passive tracer that increases linearly in regions not ventilated by surface contact and resets to zero at the surface. In CMIP6, ideal age is typically initialized during the pre-industrial control spin-up and carried through into the historical and SSP scenarios without resetting. Therefore, values reflect the cumulative ventilation history across the full simulation period. Apparent values beyond that range reflect older water masses present in the preindustrial control state, carried over into the historical run. The 400-year contour reflects water that was

last ventilated before the historical simulation began, not water aged during the historical period itself.

In this context, the absolute age in less ventilated, deep ocean may differ amongst models. Since only three models (CESM1-CAM4, CESM1-CAM5, and IPSL-CM6A-LR) have ideal ages accessible in all the simulations (SSP585, historical, mPWP, preindustrial), we present their results individually (Fig. 5 in the revised manuscript), rather than using the multi-model ensemble mean as in the original manuscript. In addition, we use potential density contours rather than absolute age contours to better depict the course of ventilation.

Comment: (2) To resolve the issues linked to the absolute age, I would suggest the authors to use the trend (time derivative) of the ideal age instead. This trend gives the rate at which a watermass is aging at any given time. This would allow the authors to compare the SSP585 and mPWP ventilation to PI ventilation directly. The authors could compute this point-wise in their zonal section, but perhaps stating a global change in ventilation would be meaningful as the authors elude to possibly increasing ocean carbon/heat uptake (which depends on other regions than SO as well).

Reply: Thank you so much for the suggestion! Please see our previous reply.

Comment: (3) L133 and L337 The authors seem to ignore sea ice melt (or is that the residual? Please add some supplementary text to explain the supplementary figures). However, I would expect that sea ice melt should be a rather large source of fresh water in SSP585 and to affect the surface density flux – Fig. 3 d-f seems to suggest that south of 65S the freshwater dominates the budget and I would expect that sea ice melt, rather than P-E would be dominating.

Reply: Yes, sea ice melt has large contribution to surface freshwater flux change south of 65°S, and hence surface density flux change there. The CMIP6 archive, however, did not provide direct output for this part such that freshwater fluxes due to sea ice melt and brine rejection are included in the residual term. We have clarified this in the Method section and supplementary figure caption in the revision.

Minor comments:

Comment: (1) In fig 1 I would suggest adding ‘negative contours are dashed’ and then in the text I would mention that Eulerian MOC is clockwise (positive) in the mean and the eddy MOC is anticlockwise (negative) in the mean. This would make it easier for the reader to digest fig 1 and realize that both positive on positive contours, and negative on negative contours, indicate strengthening. Also, I would suggest mentioning that the robust trend is the intensification of the residual MOC between 40-60S due to the intensification of the Eulerian MOC. The eddy MOC trends are hardly significant.

Reply: Following the review, we have updated the caption of Fig. 1 to clarify that “negative contours are dashed.” We have also clarified in the text that “the intensification being particularly strong between 40°S and 60°S owing primarily to wind-driven circulation change”.

Comment: (2) L104 the reader is referred to Fig 2 here, but once arriving to Fig 2 the reader is left wondering what mPWP is until reading the text to L172. Since the term mPWP is quite

prominent in Fig 2, I would suggest helping the reader a bit and spelling out the acronym in the caption to avoid unnecessary confusion.

Reply: Following the review, we have briefly introduced mPWP before the results section and spelled out the acronym in the figure caption.

Comment: (3) L86 (and thereafter) although ECCO is probably our best estimate of the SO state, I wouldn't call it 'observations', rather 'observationally constrained estimate of the Eulerian MOC...' or something similar.

Reply: Agree. Revised as suggested.

Comment: (4) L187 'a little bit' is not very precise, please rephrase.

Reply: Following the review, we have rephrased it to "slightly".

Comment: (5) L272 should probably read 'cause' instead of 'caused'

Reply: Revised as suggested.

Comment: (6) L65 'weakened' should read 'weaken'

Reply: Revised as suggested.

Comment: (7) L267-L278 brings very little added value to the manuscript. I would suggest that the authors try to tie their work to existing literature a bit tighter and give clear recommendations for future work, not just list very broad topics as is done now.

Reply: Following the review, we have revised the Discussion section and provided recommendations for future work.

Comment: (8) L338 I would ask the authors to be exact when listing what H includes, it seems to include a few things 'such as...', but the reader is left wondering if this is the complete list or if other terms contribute (and it is also unclear whether or not only the listed things are included or if something else is also included). It is also not clear what residual is in this case.

Reply: Following the review, we eliminated "such as" to make it clear what H encompasses, as well as what the residual reflects. The revised sentence reads "The term H accounts for surface heat flux components: shortwave and longwave radiation, latent and sensible heat, and residual (Supplementary Fig. 1) wherein the residual term includes heat fluxes due to frazil ice formation/melt."

Comment: (9) L359-L361 The location of mPWP data is not given. I would ask the authors to state how this data can be downloaded.

Reply: The location of mPWP data is same as the location for CMIP6 data. We have revised the Data availability to emphasize for the location for mPWP dataset in the revised manuscript.

Comment: (10) L364-365 I have to say that the code availability statement sounds odd. I would strongly recommend the authors make their code available through an online repository, such as Zenodo, with a citable DOI.

Reply: Following the review, we have updated the code available address.

Comment: (11) Reviewer #2 (Remarks on code availability): The code was not available and I have commented on it in my review

Reply: Following the review, we have updated a valid DOI in the revision. It will be publicly available upon publication. The reviewer can access it during review via the provided link:
<https://zenodo.org/records/16734876?token=eyJhbGciOiJIUzUxMiJ9.eyJpZCI6IjQ3MDNiOTJmLWE1ZjQ0NGY5Mi00OTI3LWU1OTM0YTYtMDgwYSIsImRhdGEiOiI3ZjFjMWU1Y2FmZTAzMzNjZWVhbnZlZDQ0OWU2OTI1MSJ9.hx1EylFfqXHAqKQ680Xoj4J1FxBnKB7hAIyUfQktalqDsX3ZKIn-K69z-TQIFuAu5jhdB0NMfvWKdTuvQ9rzQ>